# 120 GHz Frequency-Doubler Module Based on GaN Schottky Barrier Diode

**DOI:** 10.3390/mi13081172

**Published:** 2022-07-25

**Authors:** Honghui Liu, Zhiwen Liang, Jin Meng, Yuebo Liu, Hongyue Wang, Chaokun Yan, Zhisheng Wu, Yang Liu, Dehai Zhang, Xinqiang Wang, Baijun Zhang

**Affiliations:** 1State Key Laboratory of Optoelectronic Materials and Technologies, School of Electronics and Information Technology, Sun Yat-sen University, Guangzhou 510275, China; liuhh36@mail2.sysu.edu.cn (H.L.); liangzhw29@mail2.sysu.edu.cn (Z.L.); yck980514@163.com (C.Y.); wuzhish@mail.sysu.edu.cn (Z.W.); liuy69@mail.sysu.edu.cn (Y.L.); 2Key Laboratory of Microwave Remote Sensing, National Space Science Center, Chinese Academy of Sciences, Beijing 100190, China; mengjin@mirslab.cn (J.M.); zhangdehai@nssc.ac.cn (D.Z.); 3Science and Technology on Reliability Physics and Application of Electronic Component Laboratory, China Electronic Product Reliability and Environmental Testing Research Institute, Guangzhou 510640, China; liuyueb@foxmail.com (Y.L.); wanghongyue@pku.edu.cn (H.W.); 4State Key Laboratory for Mesoscopic Physics and Frontiers Science Center for Nano-Optoelectronics, School of Physics, Peking University, Beijing 100871, China; wangshi@pku.edu.cn

**Keywords:** frequency-doubler module, GaN, Schottky barrier diodes, terahertz

## Abstract

Traditional GaAs-based frequency multipliers still exhibit great challenges to meet the demand for solid-state high-power THz sources due to low breakdown voltage and heat dissipation of the Schottky barrier diode (SBD). In this study, a GaN SBD chain was fabricated with n^−^/n^+^-GaN structure. As a consequence, the breakdown voltage of 54.9 V at 1 μA and cut-off frequency of 587.5 GHz at zero bias were obtained. A 120 GHz frequency-doubler module based on the GaN SBD chain was designed and fabricated. When driven with 500 mW input power in a continuous wave, the output power of the frequency-doubler module was 15.1 mW at 120 GHz. Moreover, the experiments show that the frequency-doubler module can endure an input power of 2 W. In addition, it is worth noting that the SBD chain works well at an anode temperature of 337.2 °C.

## 1. Introduction

Terahertz (THz) sources are indispensable elements in the THz system. At present, the demand for solid-state high-power THz sources is expected in communications, non-destructive testing, Earth atmospheric remote sensing, and automotive radar sensors, etc. [1,2,3,4,5]. The traditional THz sources can produce high power, such as vacuum electronic devices, traveling wave tubes, quantum-cascade lasers (QCLs), and gyrotron. However, their applications are limited due to bottleneck problems, such as miniaturization, operation at room temperature, and integration with circuits [6,7]. The solid-state THz sources effectively solve these problems. They are most commonly operated by frequency multipliers of GaAs Schottky barrier diode (SBD), which realizes the nonlinear function inside waveguide. GaAs-based frequency multipliers have been achieved continuous wave (CW) sources, high stability, and reliability at room temperature [8,9,10,11,12,13,14]. The GaAs-based solid-state source can reach 2.7 THz by power amplifiers and power-combined frequency multiplier chips [6]. The output power of 250 mW was demonstrated for a short time using the 150 GHz doubler by ACST GmbH [11]. However, it is still a challenge for the GaAs-based frequency multipliers to meet requirements for high-power applications owing to the shortages of low breakdown voltage and heat dissipation. The following techniques have been adopted for the higher output power of the frequency multiplier, such as reducing heat dissipation by cooling [15], diamond substrate [16,17], film-diode technology [18,19], power-combining technology [20,21,22,23], and so on. However, these techniques result in complex THz sources architecture and higher costs.

Compared with GaAs material, GaN is a promising material for high-power THz sources due to its higher breakdown voltage and wide band gap (3.4 eV). However, there are few studies on GaN-based frequency multipliers in the THz field. Bo Zhang et al. obtained a THz source of 17.5 mW at 220 GHz by frequency tripler based on SBD of the n^−^/n^+^-GaN structure [24]. Lisen Zhang et al. obtained a THz source of 1322 mW at 150 GHz by frequency doubler based on GaN SBD [25]. Nevertheless, the output power of the frequency doubler is obtained under the pulse waveform condition, which greatly limits the application of the THz source.

In this work, a GaN SBD chain with a breakdown voltage of 54.9 V was fabricated with the n^−^/n^+^-GaN structure. A 120 GHz frequency-doubler module based on GaN-based SBD chain was designed and fabricated, which was based on the 3D model optimized by introducing parasitic parameters. The output power of the CW reached 15.1 mW at 120 GHz. In addition, the temperature distribution of the SBD chain was carried out to evaluate the reliability of the GaN-based SBD chain.

## 2. Design and Fabrication of GaN SBD Chain

In our design, a 120 GHz frequency-doubler module based on the GaN SBD chain uses the nonlinear effect of the SBD to produce the harmonics. The GaN SBD chains (parallel two chains of three series SBDs) were designed as depicted in Figure 1a. Figure 1b displays an optical micrograph of the SBD chain with the air-bridge SBD, and the overall chip is composed of six anodes with external sizes of 470 µm × 76 µm × 72 µm. An anode SBD with epitaxial information and parasitic parameters is shown in Figure 1c, which was grown by metal-organic chemical vapor deposition (MOCVD) on c-plane flat 2-inch sapphire. From bottom to top, the structure of sample consist of a 25 nm AlN, 450 nm GaN buffer layer, 2.3 μm n^+^-GaN with a Si doping concentration of 10^19^ cm^−3^, and 150 nm n^−^-GaN with a Si doping concentration of 4 × 10^17^ cm^−3^. The quality of the GaN is measured by X-ray rocking curve. The FWHM of (002) and (102) planes are 121 and 303 arcsec, respectively. The screw and edge dislocation densities are calculated as 2.94 × 10^7^ and 4.88 × 10^8^ cm^−2^, respectively [26]. GaN SBD chains were fabricated following mesa isolation by dry etching using inductively coupled plasma (ICP), cathodes metal by depositing Ti/Al/Ni/Au (15/80/20/60 nm) to n^+^-GaN, rapid thermal annealing in N_2_, Schottky metal with Ni/Au (50/100 nm), and electroplated 2.3 μm gold air-bridge. The sapphire substrate was thinned to 67 μm, and the GaN SBD chains were separated using stealth dicing technology.

The parasitic parameters reduce the conversion efficiency of the frequency multiplier, such as parasitic capacitance ($C_{pp}$ + $C_{fp}$) and series resistance ($R_{exp}$ + $R_{spreading}$ + $R_{contact}$), as shown in Figure 1c. To reduce parasitic parameters, the air-bridge GaN SBD with an anode diameter of 5 µm is adopted for a 120 GHz frequency doubler. The parasitic capacitance is reduced by deep mesa isolation and thin substrate. The values of pad-to-pad capacitance ($C_{pp}$) and finger capacitance ($C_{fp}$) are 3.0 *f*F and 1.1 *f*F, respectively. The series resistance is mainly from epitaxial resistance ($R_{exp}$). A high doping concentration of the n^+^-GaN layer can reduce the spreading resistance ($R_{spreading}$) and ohmic contact resistance ($R_{contact}$). The resistivity of the n^+^-GaN layer is 2.8 × 10^−3^ Ω cm. The specific contact resistivity is 1.41 × 10^−5^ Ω cm^2^.

We measured the capacitance-voltage (*C*-*V*) and current-voltage (*I*-*V*) characteristics by the Keysight B1505A and Keithley 4200-SCS semiconductor parameter analyzer at room temperature (RT), respectively. The tests were performed at room temperature. Figure 2a,b shows experimental results of GaN SBDs with three anodes. It can be seen that both GaN SBDs with and without alloying exhibit good Schottky diode properties. There is no essential difference for the *I*-*V* and *C*-*V* curves between room temperature and after alloying of 400 °C in O_2_ ambient. The important parameters of GaN SBDs are almost unchanged after alloying at 400 °C in O_2_ ambient. The result reflects that GaN SBDs have relatively stable characteristics over this temperature range.

The expression of junction capacitance ($C_{j}$($V_{j}$)) can be seen in Equation (1):(1)$$C_{j}\left(V_{j}\right)=\frac{dQ_{j}}{dV_{j}}\;=\;\frac{C_{j}\left(0\right)}{\sqrt{1-V_{j}/V_{bi}}}$$
(2)$$C_{j}\left(0\right)\;=\;\;A\sqrt{\frac{q\varepsilon_{0}\varepsilon_{r}N_{d}}{2V_{bi}}}$$
in which $V_{j}$ is junction voltage; $Q_{j}$ is junction charge; $C_{j}\left(0\right)$ is zero bias capacitance; $V_{bi}$ is applied voltage bias; A is anode area; q is the elementary charge; $\varepsilon_{0}$ is vacuum dielectric constant; $\varepsilon_{r}$ is the dielectric constant of GaN; and $N_{d}$ is carrier concentration. From Figure 2a, the capacitance decreases from 10.3 *f*F to 4.3 *f*F; the trend of change for junction capacitance is basically consistent with the theoretical model.

The turn-on voltage of ~1.74 V at 1 µA, the series resistance ($R_{s}$) of 26.3 Ω, the ideal factor of 1.24, and barrier voltage of 0.78 V are extracted from forward *I*-*V* data. The breakdown voltage of 54.9 V at 1 μA is displayed from reverse *I*-*V* data, as shown in Figure 2b. The cut-off frequency ($f_{c}$) limits the operating frequency in the frequency multiplier, which depends on $R_{s}$ and $C_{j}$ as follows:(3)$$f_{c}=\frac{1}{2\times \pi \times R_{s}\times C_{j}}$$$R_{s}$ and $C_{j}$ are set as 10.3 fF and 26.3 Ω, and the *f_c_* of GaN SBD is calculated to be 587.5 GHz at zero bias.

In order to better describe the material properties of GaN and GaAs, the main parameters of the GaAs and GaN SBD with an anode are summarized in Table 1. Compared with GaAs SBD, GaN SBD exhibits higher breakdown voltage due to a wide band gap. Therefore, GaN-based frequency multipliers can withstand higher input power. However, the large series resistance hinders the conversion efficiency of the frequency multipliers due to the low electron mobility.

The main parameters of the GaAs and GaN SBD with one anode are summarized in Table 1. Compared with GaAs SBD, GaN SBD exhibits higher breakdown voltage due to a wide band gap. Therefore, GaN-based frequency multipliers can withstand higher input power. However, the large series resistance hinders the conversion efficiency of the frequency doubler due to the low electron mobility.

The heat dissipation of the frequency doubler greatly reduces the conversion efficiency [16,17,18]. Nevertheless, it is difficult to obtain the temperature distribution of the SBD chain due to the sealed waveguide and small-sized anode. We firstly measured the temperature distribution of the SBD chain using a thermoreflectance imaging system, as shown in Figure 3. Under reverse bias, the anode temperature (*T_Anode_*) of the SBD chain was kept at room temperature due to a low leakage current. Under forward bias, the *T_Anode_* rapidly rises with the increase of the SBD chain current. The heat generation of the SBD chain was mainly concentrated in the anode area, which matched the distribution of $R_{s}$. Furthermore, the GaN SBD chain can continue to operate with a current of 92 mA at 337.2 °C.

## 3. Frequency-Doubler Module

### 3.1. Design of 120 GHz Frequency Doubler

The 120 GHz frequency-doubler module is composed of four parts: the input waveguide, GaN SBD chain, the output matching, and the output waveguide. The 3D model is demonstrated in Figure 4. The inset shows the details of the GaN SBD chain. The signal (TE_10_ mode at *f_0_*) was coupled through a waveguide-microstrip structure, and the size and the location of the waveguide were optimized to reduce the return loss of the input port. The output wave (TEM mode at 2*f_0_*) passed through the output matching and couples into the output probe. The direct current (DC) filter, which is H-shaped, reduced the loss at the bias circuit.

A global field-circuit method is applied to design the 120 GHz frequency doubler, which is divided into two parts: (1) a linear network is analyzed by Ansys High-Frequency Structure Simulator (HFSS); (2) the parasitic parameters and nonlinear behavior of GaN SBD are solved by Agilent Advanced Design Simulator (ADS).

The equivalent models of 120 GHz frequency doubler are shown in Figure 5. There is a one-to-one matching relationship between the elements of circuits and models. The circuit is designed to obtain the optimal impedance of the GaN SBD, which is based on the pull between source and load at the fundamental and second harmonic. The results show that the optimal impedance values of the source and load are 43-j16 Ω and 17-j11 Ω, respectively. The signal of 60 GHz is inputted in the waveguide microstrip, and the equivalent models of the 120 GHz frequency doubler achieve the return loss (S11) of below −15 dB. The models of ADS (six diodes of GaN SBD chain) are connected between the input waveguide and the output matching. The second harmonic of 120 GHz passes the output matching and couples into the output probe. The DC filter and microstrip-waveguide transition are designed in the output structure, and the DC bias is applied to the GaN SBD chain via the output microstrip waveguide.

The 120 GHz frequency doubler achieves optimal output conversion by optimizing DC bias. Figure 6 shows the simulation results about output power and conversion efficiency vs. frequency of 120 GHz frequency doubler. When driven with input power of 200 mW, the output power reach a peak value at 120 GHz, and the peak output power of reach 13.5 mW with a conversion efficiency of 6.8%.

### 3.2. Fabrication of 120 GHz Frequency Doubler

The fabrication of 120 GHz frequency-doubler module includes the E-plane split-waveguide copper block, waveguide-microstrip structure, and GaN SBD chain. The waveguide-microstrip structure was a deposited 2 μm gold microstrip on 50 μm quartz wafer by electron beam evaporation. The waveguide-microstrip structure was assembled in the E-plane split-waveguide copper block. The GaN SBD chain was assembled on the quartz circuit of output matching by using the conductive adhesive. Assembly details of the 120 GHz frequency-doubler module are shown in Figure 7a. The sub-miniature version A (SMA) coaxial radio frequency (RF) connector and the main transmission circuit were connected by gold wire bonding. Finally, the 120 GHz frequency doubler was fabricated by combining two E-plane split-waveguide copper blocks, as shown in Figure 7b.

## 4. Results and Discussion

The input power source of drive 120 GHz frequency-doubler module is a 6× frequency multiplier, which exhibits a input frequency from 50 to 75 GHz. The Agilent analog signal generator E8257D provided the input signal, and the output power of the 120 GHz frequency-doubler module was detected by the PM4 power meter. A voltage of −8.4 V was applied to the anode of the GaN SBD chain by SMA, which can perform a RF swing of high input power and optimal conversion efficiency. A photograph of the test site for the 120 GHz frequency-doubler module is shown in Figure 8.

When driven with 200 mW input power in CW, the output power reached a peak value at 120 GHz, as shown in Figure 9a. There is an agreement between the simulations and measurements for both conversion efficiency and output power, but it is lower than the simulated results. The reason for the difference is mainly due to the loss of the waveguide and assembly error. Figure 9b shows peak output power and conversion efficiency of 120 GHz vs. different input power. The output power of the CW reaches 15.1 mW at 120 GHz with a conversion efficiency of 3%. The lower conversion efficiency is mainly due to the large $R_{s}$. When the driving power is higher than 0.5 W, the conversion efficiency decreases due to the heat generated and accumulated in the GaN-based SBD chain. The experiments show that the 120 GHz frequency-doubler module can endure an input power of 2 W due to higher breakdown voltage.

Table 2 summarizes the comparison of the key performance between our 120 GHz frequency doubler and other similar solid-state frequency multipliers in a recent report. Compared to GaAs frequency multipliers, the GaN frequency multipliers show much higher power-handling ability. The frequency multipliers of the pulsed output can support higher output power than the CW output due to less heat generation. In the space of waveguide, more anodes can enhance the higher output power. In this work, the 120 GHz frequency doubler generated 15.1 mW CW output power with a conversion efficiency of 3%. By comparing the GaAs-based frequency multiplier (power-handling capability of 0.5 W), our frequency doubler endured an input power of 2 W due to a breakdown voltage of 54.9 V at 1 µA. In addition, GaN SBD on a sapphire substrate has a very low cost compared with GaAs SBD and GaN SBD on SiC.

## 5. Conclusions

In this work, a solid-state 120 GHz frequency-doubler module of the GaN SBD chain was designed and fabricated with the n^−^/n^+^-GaN substrate. The GaN SBD chain has a breakdown voltage of 54.9 V at 1 μA, a series resistance of 26.3 Ω, and a cut-off frequency of 587.5 GHz at zero bias. The frequency-doubler module generated 15.1 mW output power at 120 GHz, which was driven with 500 mW input power of the CW. The frequency-doubler module can endure an input power of 2 W due to higher breakdown voltage. Moreover, the temperature distribution of the SBD chain was measured for the first time. The GaN SBD chain works successfully at 337.2 °C. In the following research, we focus on the influencing factor of parasitic resistance and substrate loss, which can improve the conversion efficiency and output power.

## Figures and Tables

**Figure 1 micromachines-13-01172-f001:**
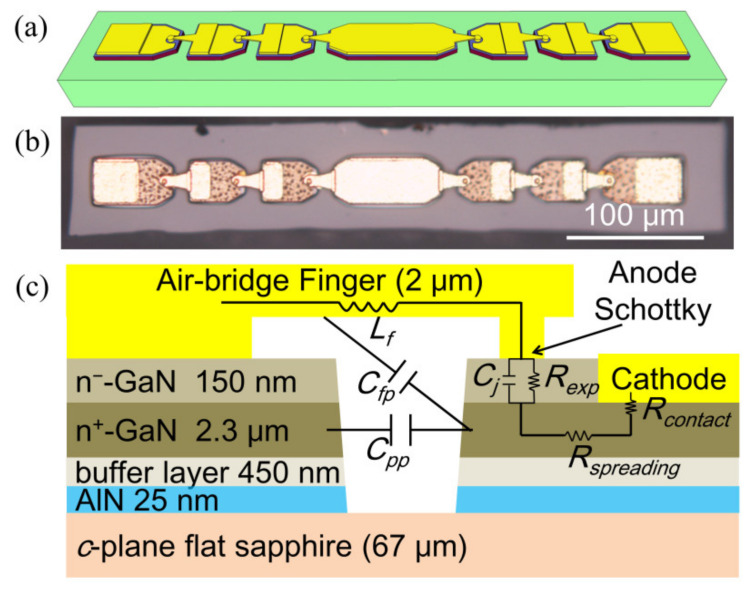
(**a**) The 3D structure of the GaN SBD chain. (**b**) The optical micrograph of the GaN SBD chain. (**c**) The schematic cross-section of air-bridge SBD with epitaxial information.

**Figure 2 micromachines-13-01172-f002:**
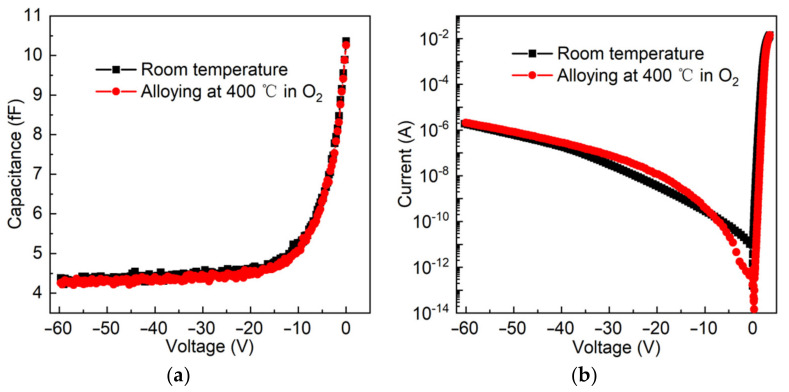
Characteristics of the fabricated GaN SBD at room temperature and after alloying of 400 °C in O_2_. (**a**) Measured *C*-*V* curves. (**b**) Measured *I*-*V* curves.

**Figure 3 micromachines-13-01172-f003:**
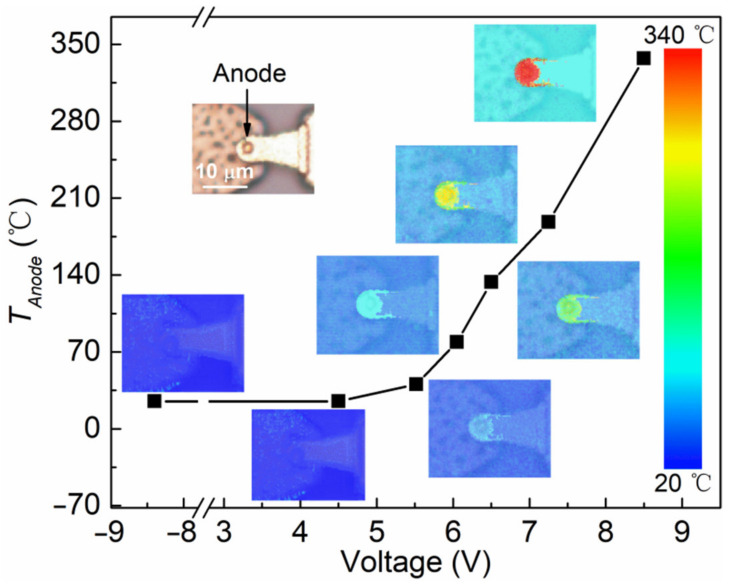
Anode of GaN SBD chain temperature distribution at different bias voltages.

**Figure 4 micromachines-13-01172-f004:**
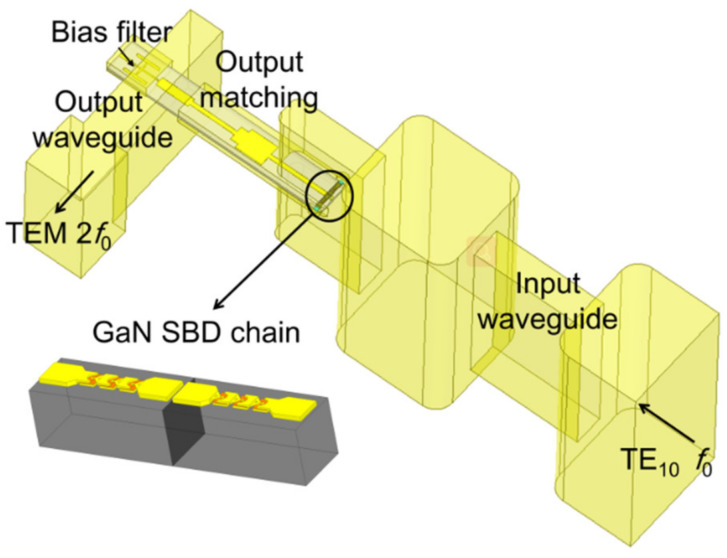
The overall 3D model of 120 GHz frequency doubler in HFSS. The inset shows the GaN SBD chain in the frequency doubler.

**Figure 5 micromachines-13-01172-f005:**
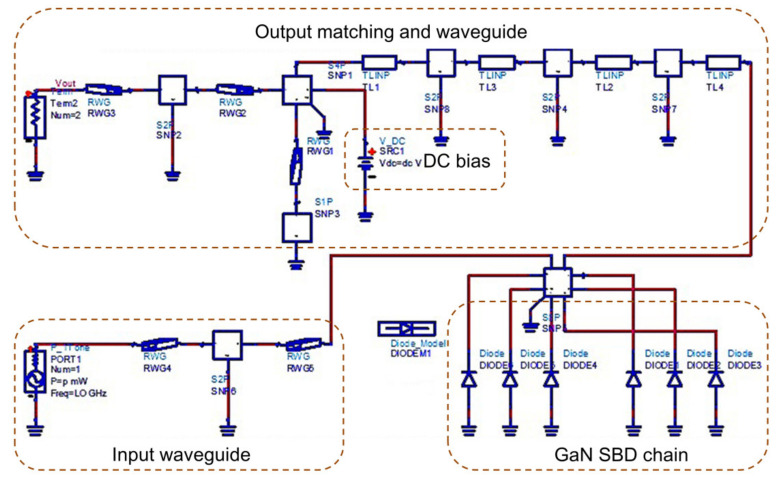
Design of the 120 GHz monolithic integrated doubler.

**Figure 6 micromachines-13-01172-f006:**
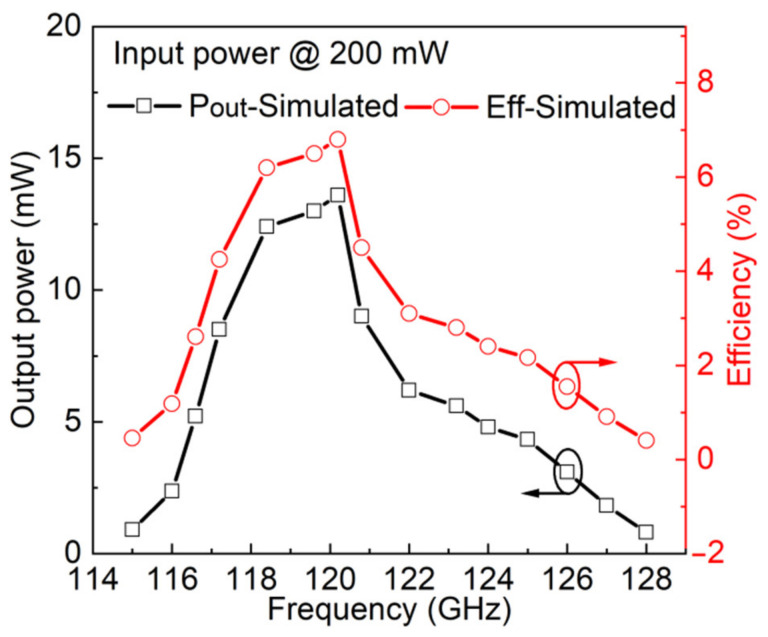
Simulated output power and efficiency of 120 GHz frequency doubler at input power of 200 mW.

**Figure 7 micromachines-13-01172-f007:**
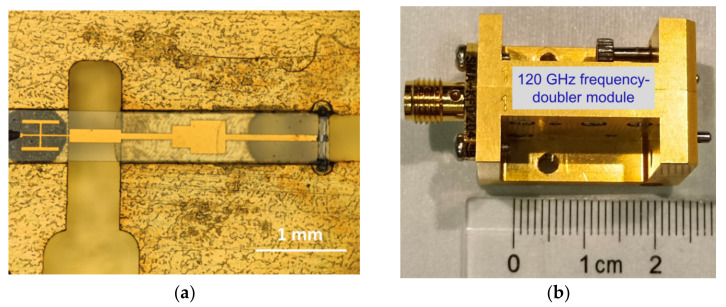
(**a**) The optical micrograph of 120 GHz frequency doubler imbedded in the split-waveguide block. (**b**) Overall photograph of the 120 GHz frequency-doubler module.

**Figure 8 micromachines-13-01172-f008:**
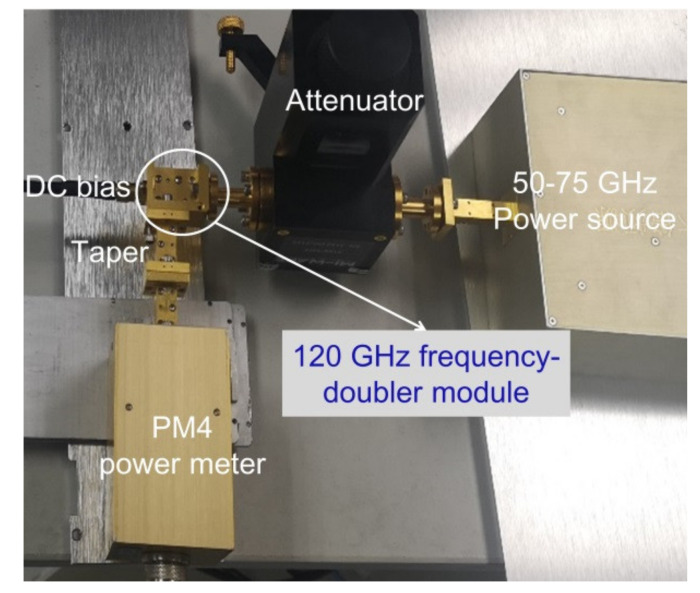
Photograph of test site for the 120 GHz frequency-doubler module.

**Figure 9 micromachines-13-01172-f009:**
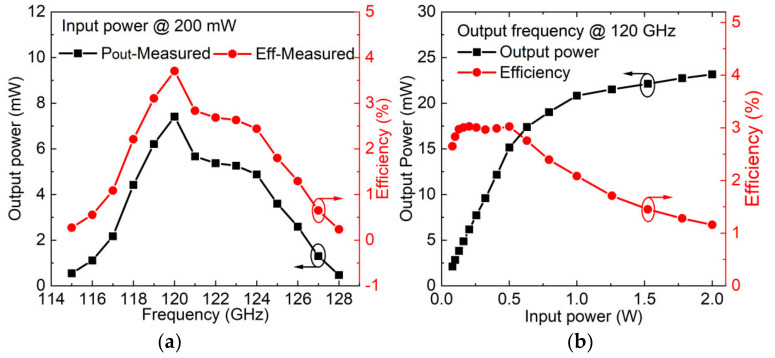
Measurement results of 120 GHz frequency-doubler module. (**a**) Measured output power and efficiency. (**b**) Measured output power and efficiency.

**Table 1 micromachines-13-01172-t001:** Comparison of main parameters of the GaAs and GaN SBD with one anode.

Materials	GaAs SBD [27]	GaN SBD
Thickness of epitaxial layer (nm)	300	150
Doping concentration of epitaxial layer (cm^−^^3^)	2 × 10^−17^	4 × 10^−17^
Series resistance (Ω)	3	8.8
Zero bias capacitance (*f*F)	40	30.3
Barrier voltage (V)	0.9	0.78
Breakdown voltage (V at µA)	10	18.3
Band gap width (eV)	1.42	3.4
Ideal factor	1.2	1.24
Electron mobility (cm^2^V^−^^1^s^−^^1^)	8000	900

**Table 2 micromachines-13-01172-t002:** Performance comparison of similar frequency multipliers.

Ref.	Diode Material	Multiplying Factor	Frequency Band (GHz)	Input Power (mW)	Output Power (mW)	Conversion Efficiency	Number of Anodes
[9]	GaAs	×2	170–200	500	125	26%	6
[12]	GaAs	×2	184–212	180	54	30%	6
[14]	GaAs	×2	190–198	260	20	7.7%	6
[27]	GaAs	×2	190–235	89	21.4	24%	6
[28]	GaAs	×2	135–190	174	13	7.5%	4
[29]	GaAs	×3	210–218	300	10.5	3.5%	6
[24]	GaN	×3	200–220	1100	17.5	1.6%	8
[30] *	GaN	×2	175–185	2140	244	11.4%	8
This work	GaN	×2	117–125	500	15.1	3%	6

* Multipliers of the pulsed output.

## Data Availability

The data that support the finding of this study are available from the corresponding author upon reasonable request.

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
