# Peer review of "120 GHz Frequency-Doubler Module Based on GaN Schottky Barrier Diode"

_micromachines, 2022, doi:10.3390/mi13081172_

Round 1

Reviewer 1 Report

This manuscript deliver significant results. I recommend for Micromachines journal. There are some minor suggestions/comments:

1)     Please remove unnecessary “The” before, Page 2, Line 55.

2)     Page 2, Why breakdown voltage of the GaN SBDs is relatively low (about 54.9 V). The breakdown voltage > 100 V is reported in the literature for vertical GaN SBD structures.  

3)     Page 3, “This also reflects the high-temperature stability and reliability of GaN SBDs”. Note that, the characteristics are measured after 400 °C in O2 ambient. In fact, this temperature is not high, as because the high-temperature reliability studies are generally carried out at elevated temperatures (> 800 °C) for longer time periods. Here alloying of 400 °C. Hence, you may consider to rewrite this sentence. 

Author Response

We would like to thank you for your careful reading and helpful comments.

Reviewer 2 Report

The authors demonstrate GaN-based frequency doubler with 500 mW input power and 15.1 mW at 120 GHz. The temperature distribution of the device is evaluated, which indicates high-temperature stability and reliability. I found the results are timely interested and the manuscript is well-organized with sufficient details. I suggest this manuscript can be published in Micromachines.

Some minor suggestions:

1. unlike arsenide material system which is mature, the GaN quality varies depending on the growth conditions. The authors should provide an estimate of the dislocation density

2. there are obvious writing mistakes such as Page 2, line 55 as well as others, the author should check the manuscript thoroughly

Author Response

(The authors gave the same response as above.)
